# FSPO: FEW-SHOT OPTIMIZATION OF SYNTHETIC PREFERENCES PERSONALIZES TO REAL USERS

**Anikait Singh**[1,*]**, Sheryl Hsu**[1,*]**, Kyle Hsu**[1]**, Eric Mitchell**[1,3]
**Stefano Ermon**[1]**, Tatsunori Hashimoto**[1]**, Archit Sharma**[1,2,*]**, Chelsea Finn**[1,*]

[1]Stanford University [2]Google DeepMind [3]OpenAI [*]Equal Contribution/Advising

## ABSTRACT

Effective personalization of LLMs is critical for a broad range of user-interfacing applications such as virtual assistants and content curation. Inspired by the strong in-context capabilities of LLMs, we propose few-shot preference optimization (FSPO), an algorithm for LLM personalization that reframes reward modeling as a meta-learning problem. Under FSPO, an LLM learns to quickly infer a personalized reward function for a user via a few labeled preferences. FSPO also utilizes user description rationalization (RAT) to encourage better reward modeling and instruction following, recovering performance with the oracle user description. Since real-world preference data is challenging to collect at scale, we propose careful design choices to construct synthetic preference datasets for personalization, generating over 1M synthetic personalized preferences using publicly available LLMs. To successfully transfer from synthetic data to real users, we find it crucial for the data to exhibit both high diversity and coherent, self-consistent structure. We evaluate FSPO on personalized open-ended generation for up to 1,500 synthetic users across three domains: movie reviews, education, and open-ended question answering. We also run a controlled human study. Overall, FSPO achieves an 87% Alpaca Eval winrate in generating responses that are personalized to synthetic users and a 70% winrate with real human users in open-ended question answering.

## 1 INTRODUCTION

As large language models (LLMs) increasingly interact with a diverse user base, it becomes important for models to generate responses that align with individual user preferences. People exhibit a wide range of preferences and beliefs shaped by their cultural background, personal experience, and individual values. These diverse preferences are present in human-annotated preference datasets; however, current preferences optimization techniques like reinforcement learning from human feedback (RLHF) largely focus on optimizing a *single* model based on preferences aggregated over the entire population. This approach may neglect minority viewpoints, embed systematic biases into the model, and ultimately lead to worse performance compared to personalized models. Can we create language models that can adaptively align with the personal preferences of each user instead of the aggregated preferences of all users?

Addressing this challenge requires a shift from modeling a singular aggregate reward function to modeling a distribution of reward functions that captures the diversity of human preferences [41, 18]. By doing so, we can enable personalization in language models, allowing them to generate a wide range of responses tailored to individual subpopulations. This approach not only enhances user satisfaction but also promotes inclusivity by acknowledging and respecting the varied perspectives that exist within any user base. Despite this problem's importance, to our knowledge LLM personalization has yet to be achieved for open-ended question answering with real users.

In this paper, we introduce few-shot preference optimization (FSPO), a novel framework designed to model diverse subpopulations in preference datasets to elicit personalization in language models for open-ended question answering. At a high level, FSPO leverages in-context learning to adapt to new subpopulations. This adaptability is crucial for practical applications, where user preferences can be dynamic and multifaceted. Inspired by past work on black-box meta-learning for language

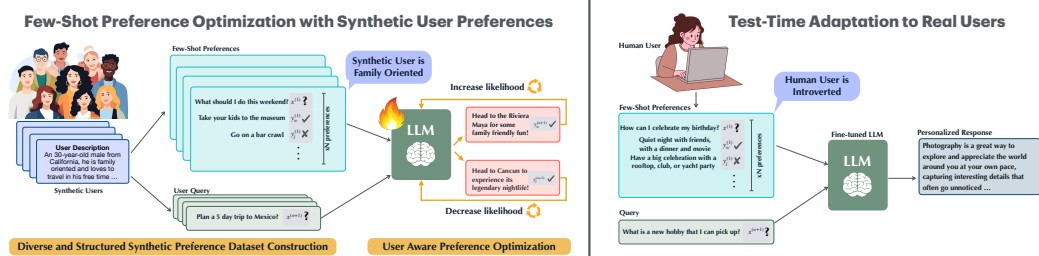

Figure 1: **Overview of FSPO.** $N$ previously collected preferences are fed into the LLM along with the current query, allowing the LLM to personalize its response to the query using the past preferences. Furthermore, user description rationalization (e.g Synthetic user is family-oriented) is utilized to predict details about a user from their preferences in natural language, aiding reward modeling and text generation.

modeling [6, 28, 51], we fine-tune the model in a meta-learning setup using preference-learning objectives such as IPO [12]. To further improve personalized generation, we additionally propose user description rationalization (RAT), which allows the model to leverage additional inference-time compute for better reward modeling and instruction following.

Learning a model that effectively personalizes to real people requires training on a realistic, user-stratified preference dataset. One natural approach to consider is to curate such data from humans, but this is difficult and time-consuming. Instead, we propose instantiating this dataset synthetically, and present careful design decisions inspired from the meta-learning literature [16, 50] to generate a dataset that is both diverse and structured.

To evaluate the efficacy of our approach, we construct a set of three semi-realistic domains to study personalization: (1) **Reviews**, studying the generation ability of models for reviews of movies, TV shows, and books that are consistent with a user's writing style, (2) **Explain Like I'm X (ELIX)**: studying the generation ability of models for responses that are consistent with a user's education level, and (3) **Roleplay**: studying the generation ability of models for responses that are consistent with a user's description, with effective transferability to a real human-study. Here we find that FSPO outperforms an unpersonalized model on average by 87%. We additionally perform a controlled human study showcasing a winrate of 70% of FSPO over unpersonalized models.

By addressing limitations of existing reward modeling techniques, our work paves the way for more inclusive and personalized LLMs. We believe that FSPO represents a significant step toward models that better serve the needs of all users, respecting the rich diversity of human preferences.

## 2 RELATED WORK

**Personalized learning of preferences.** Prior research has explored personalization through various methods. One approach is distributional alignment, which focuses on matching model outputs to broad target distributions rather than tailoring them to individual user preferences. For example, some prior work have concentrated on aligning model-generated distributions with desired statistical properties [40, 26, 27], yet they do not explicitly optimize for individual preference adaptation. Another strategy involves explicitly modeling a distribution of rewards [21, 34]. However, these methods suffer from sample inefficiency during both training and inference [36, 12]. Additionally, these approaches have limited evaluations: Lee et al. [21] focuses solely on reward modeling, while Poddar et al. [34] tests with a very limited number of artificial users (e.g helpfulness user and honest user). Other works have investigated personalization in multiple-choice questions, such as GPO [54]. Although effective in structured survey settings, these methods have not been validated for open-ended personalization tasks. Similarly, Shaikh et al. [39] explores personalization via explicit human corrections, but relying on such corrections is expensive and often impractical to scale. Finally, several datasets exist for personalization, such as Prism [19] and Persona Bench [5]. Neither of these datasets demonstrate that policies trained on these benchmarks lead to effective personalization. Unlike these prior works which study personalization based off of human values and controversial questions, we instead study more general questions that a user may ask.

**Algorithms for preference learning.** LLMs are typically fine-tuned via supervised next-token prediction on high-quality responses and later refined with human preference data [4, 33]. This

process can use on-policy reinforcement learning methods like REINFORCE [42] or PPO [38], which optimize a reward model with a KL constraint. Alternatively, supervised fine-tuning may be applied to a curated subset of preferred responses [11] or iteratively to preferred completions as in ReST [15]. Other methods, such as DPO [36], IPO [12], and KTO [8], learn directly from human preferences without an explicit reward model, with recent work exploring iterative preference modeling applications [52].

**Black-box meta-learning.** FSPO is an instance of black-box meta-learning, which has been studied in a wide range of domains spanning image classification [37, 29], language modeling [6, 28, 51], and reinforcement learning [9, 45]. Black-box meta-learning is characterized by the processing of task contexts and queries using generic sequence operations like recurrence or self-attention, instead of specifically designed adaptation mechanisms.

## 3 PRELIMINARIES AND NOTATION

Preference fine-tuning algorithms, such as reinforcement learning from human feedback (RLHF) and direct preference optimization (DPO), typically involve two main stages [33, 32]: supervised fine-tuning (SFT) and preference optimization (DPO/RLHF). First, a pre-trained model is fine-tuned on high-quality data from the target task using SFT. This process produces a reference model, denoted as $\pi_{\text{ref}}$. The purpose of this stage is to bring the responses from a particular domain in distribution with supervised learning. To further refine $\pi_{\text{ref}}$ according to human preferences, a preference dataset $\mathcal{D}_{\text{pref}} = \{(\mathbf{x}^{(i)}, \mathbf{y}_w^{(i)}, \mathbf{y}_l^{(i)})\}$ is collected. In this dataset, $\mathbf{x}^{(i)}$ represents a prompt or input context, $\mathbf{y}_w^{(i)}$ is the preferred response, and $\mathbf{y}_l^{(i)}$ is the less preferred response. These responses are typically sampled from the output distribution of $\pi_{\text{ref}}$ and are labeled based on human feedback.

Most fine-tuning pipelines assume the existence of an underlying reward function $r^*(\mathbf{x}, \cdot)$ that quantifies the quality of responses. A common approach to modeling human preferences is the Bradley-Terry (BT) model [2], which expresses the probability of preferring response $\mathbf{y}_1$ over $\mathbf{y}_2$, given a prompt $\mathbf{x}$, as:

$$p^*(\mathbf{y}_1 \succ \mathbf{y}_2 \mid \mathbf{x}) = \frac{e^{r^*(\mathbf{x}, \mathbf{y}_1)}}{e^{r^*(\mathbf{x}, \mathbf{y}_1)} + e^{r^*(\mathbf{x}, \mathbf{y}_2)}} \tag{1}$$

Here, $p^*(\mathbf{y}_1 \succ \mathbf{y}_2 \mid \mathbf{x})$ denotes the probability that $\mathbf{y}_1$ is preferred over $\mathbf{y}_2$ given $\mathbf{x}$.

The objective of preference fine-tuning is to optimize the policy $\pi_\theta$ to maximize the expected reward $r^*$. However, directly optimizing $r^*$ is often impractical due to model limitations or noise in reward estimation. Therefore, a reward model $r_\phi$ is trained to approximate $r^*$. To prevent the fine-tuned policy $\pi_\theta$ from deviating excessively from the reference model $\pi_{\text{ref}}$, a Kullback-Leibler (KL) divergence constraint is imposed. This leads to the following fine-tuning objective:

$$\max_\pi \ \mathbb{E}[r^*(x, y)] - \beta \, D_{\text{KL}}(\pi \parallel \pi_{\text{ref}}) \tag{2}$$

In this equation, the regularization term weighted by $\beta$ controls how much $\pi_\theta$ diverges from $\pi_{\text{ref}}$, based on the reverse KL divergence constraint. This constraint ensures that the updated policy remains close to the reference model while improving according to the reward function.

**Reward model training.** To fine-tune the large language model (LLM) policy $\pi_\theta(\mathbf{y} \mid \mathbf{x})$, the Bradley-Terry framework allows for either explicitly learning a reward model $r_\phi(\mathbf{x}, \mathbf{y})$ or directly optimizing preferences. Explicit reward models are trained using the following classification objective:

$$\max_\phi \ \mathbb{E}_{\mathcal{D}_{\text{pref}}} \left[\log \sigma \left(r_\phi(\mathbf{x}, \mathbf{y}_w) - r_\phi(\mathbf{x}, \mathbf{y}_l)\right)\right] \tag{3}$$

where $\sigma$ is the logistic function, used to map the difference in rewards to a probability. Alternatively, contrastive learning objectives such as Direct Preference Optimization [36] and Implicit Preference Optimization [12] utilize the policy's log-likelihood $\log \pi_\theta(\mathbf{y} \mid \mathbf{x})$ as an implicit reward:

$$r_\theta(\mathbf{x}, \mathbf{y}) = \beta \log \left(\pi_\theta(\mathbf{y} \mid \mathbf{x}) / \pi_{\text{ref}}(\mathbf{y} \mid \mathbf{x})\right) \tag{4}$$

This approach leverages the policy's log probabilities to represent rewards, thereby simplifying the reward learning process.

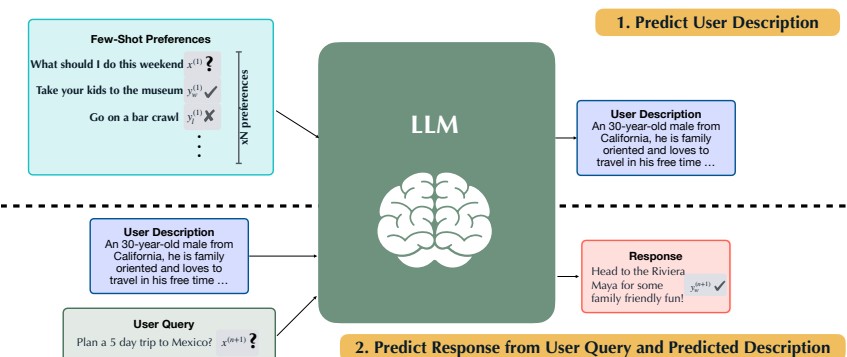

Figure 2: **User Description Rationalization (RAT).** Prediction is a two-stage process: first predicting a (synthetic) user description from the few-shot preferences and next predicting the response. The model is fine-tuned with a reward of how close the generated user description is to the gold user description.

## 4  THE FEW-SHOT PREFERENCE OPTIMIZATION (FSPO) FRAMEWORK

**Personalization as a meta-learning problem.** Generally, for fine-tuning a model with RLHF a preference dataset of the form: $\mathcal{D}_{\text{pref}} = \{(\mathbf{x}^{(i)}, \mathbf{y}_w^{(i)}, \mathbf{y}_l^{(i)})\}$ is collected, where $x$ is a prompt, $y_w$ is a preferred response, and $y_l$ is a dispreferred response. Here, preferences from different users are aggregated to learn the preferences over a population. However, through this aggregation, individual user preferences are marginalized, leading to the model losing personalized values or beliefs due to population-based preference learning and RLHF algorithms such as DPO as seen in prior work [40].

How can we incorporate user information when learning from preference datasets? In this work, we have a weak requirement to collect scorer-ids $\mathbf{S}^{(i)}$ of each user for differentiating users that have labeled preferences in our dataset: $\mathcal{D}_{\text{pref}} = \{(\mathbf{x}^{(i)}, \mathbf{y}_w^{(i)}, \mathbf{y}_l^{(i)}, \mathbf{S}^{(i)})\}$. Now consider each user as a task instance, where the objective is to learn an effective reward function for that user using the user's set of preferences. This can be naturally instantiated as a black-box meta-learning objective, where meta-learning is done over users (also referred to as a task in meta-learning). Meta-learning should enable rapid personalization, i.e. adaptability to new users with just a few preferences.

More formally, consider that each unique user $\mathcal{S}^{(i)}$'s reward function is characterized by a set of preferences with prompt and responses $(x, y_1, y_2)$, and preference label $c$ (indicating if $y_1 \succ y_2$ or $y_1 \prec y_2$). Given a distribution over users $\mathcal{S} = P(\mathcal{S}^{(i)})$, a meta-learning objective can be derived to minimize its expected loss with respect to $\theta$ as:

$$\min_\theta \mathbb{E}_{\mathcal{S}^{(i)} \sim \mathcal{S}} \left[ \mathbb{E}_{(x, y_1, y_2, c) \sim \mathcal{D}_i, \{(x, y_1, y_2, c)\}_1^N \sim \mathcal{D}_i} \left[ \mathcal{L}_{pref}^\theta \left( x, y_1, y_2, c | \{(x, y_1, y_2, c)\}_1^N \right) \right] \right] \quad (5)$$

where $D_i$ is a distribution over preference tuples $(x, y_1, y_2, c)$ for each user $S^{(i)}$, and $\mathcal{L}_{pref}^\theta$ is a preference learning objective such as DPO [36] or IPO [12]:

$$\mathcal{L}_{pref}^\theta = ||h_{\pi_\theta}^{y_w, y_l} - (2\beta)^{-1}||_2^2, \quad h_{\pi_\theta}^{y_w, y_l} = \log \frac{\pi_\theta(y_w|x)}{\pi_{ref}(y_w|x)} - \log \frac{\pi_\theta(y_l|x)}{\pi_{ref}(y_l|x)} \quad (6)$$

where $y_w$ and $y_l$ are the preferred and dispreferred responses (respectively) according to the responses $y_1, y_2$ and class label $c$ in the preference dataset.

Following black-box meta-learning approaches, FSPO receives as input a sequence of preferences $D_i^{fewshot} \sim D_i$ from a User $S^{(i)}$. This is followed by an unlabeled, held-out preference $(x, y_1, y_2) \sim \mathcal{D}_i \backslash \mathcal{D}_i^{fewshot}$ for which it outputs its prediction $c$. To make preferences compatible with a pre-trained language model, a few-shot prompt is constructed, comprising of preferences from a user and the held-out query as seen in Figure 1. This construction has an added benefit of leveraging a pretrained language model's capabilities for few-shot conditioning [3], which can enable some amount of steerage/personalization. This prediction $c$ is implicitly learned by a preference optimization algorithm such as DPO [36], which parameterizes the reward model as $\beta \frac{\log \pi_\theta(y|x)}{\log \pi_{ref}(y|x)}$. This parameterization enables us to leverage the advantages of preference optimization algorithms such as eliminating

policy learning instabilities and computational burden of on-policy sampling, learning an effective model with a simple classification objective.

**User description rationalization (RAT).** If provided with a description of the user (potentially synthetically generated), FSPO can be converted to a two-step prediction problem as seen in Figure 2. In the first step, conditioned on user few-shot preferences, the user description is generated, then conditioned on the prompt, few-shot preferences, and generated user description, a response can then be generated (Example in Appendix A.2.1). This prediction of the user description is an interpretable summarization of the fewshot preferences and a better representation to condition on for response generation. Similar to the rationale generated in Zhang et al. [53] for verifiers, the RAT prediction can be viewed as using additional inference-compute for better reward modeling. Additionally, this formulation leverages the instruction following ability of LLMs [33] for response generation.

This rationalization procedure is expert-guided, fine-tuned with preference pairs over on-policy samples of a user description, where a preferred user description is one that is semantically closer to the ground-truth user description, conditioned on few-shot examples from the user. This benefits the optimization procedure twofold by (1) leveraging additional inference-compute for better reward modeling and (2) utilizing the instruction-following ability of LLMs for response generation. The instantiation of this rationalization optimization is unique, fundamentally different from COT approaches present in reasoning tasks, which use rule-based rewards to train Long-COT models for math and code reasoning. For an open-ended task, such verifiers do not exist and thus requires a different instantiation. We additionally show in Appendix A.2.1, a sample persona generated with RAT and that it qualitatively matches the underlying held-out user description, showing the efficacy of the procedure to recover characteristics about an *unseen user*.

**User representation through preference labels**. From an information-theoretic perspective, the few-shot binary preferences can be seen as a $N$-bit representation of the user, representing up to $2^N$ different personas or reward functions. There are several ways to represent users: surveys, chat histories, or other forms of interaction that reveal hidden preferences. We restrict our study to such a $N$-bit user representation, as such a constrained representation can improve the performance when transferring reward models learned on synthetic personalities to real users. We defer the study of less constrained user representations to future work.

---

**Algorithm 1** Overview of Few-Shot preference optimization (FSPO).

---

1: **Input:** For each unique user $\mathcal{S}^{(i)}$, a dataset of preferences $\mathcal{D} := (x, y_1, y_2, c)_i$, and optionally user description $y^+_{\mathcal{S}^{(i)}}, y^-_{\mathcal{S}^{(i)}}$ for RAT ($+$ is preferred user description and $-$ is dispreffered user description given gold user description $y^*_{\mathcal{S}^{(i)}}$), $\forall i$
2: **Output:** Learned policy $\pi_\theta$
3: **while** not done **do**
4:  Sample training user $\mathcal{S}^{(i)}$ (or minibatch)
5:  Sample a subset of preferences from the user $\mathcal{D}^{fewshot}_i \sim \mathcal{D}_i$
6:  Sample held-out preference examples $D^{heldout}_i \sim \mathcal{D}_i \backslash \mathcal{D}^{fewshot}_i$
7:  **if** RAT **then**
8:   Use Eq. (5) and Eq. (6) to predict the loss on the user descriptions $y^+_{\mathcal{S}^{(i)}}$ and $y^-_{\mathcal{S}^{(i)}}$.
9:  **end if**
10:  Conditioning on $\mathcal{D}^{fewshot}_i$ (optionally $y_{\mathcal{S}^{(i)}}$), use Eq. (5) and Eq. (6) to predict the loss on the held-out preference example $D^{heldout}_i$
11:  Update learner parameters $\theta$, using gradient of loss on $D^{heldout}_i$
12: **end while**
13: **Return** $\pi_\theta$

---

We summarize FSPO in Algorithm 1. Next, we will discuss domains to study FSPO.

## 5 CONSTRUCTING A TESTBED FOR PERSONALIZATION

To study personalization with FSPO we construct a benchmark across 3 domains ranging from generating personalized movie reviews (**Reviews**), generating personalized responses based off a

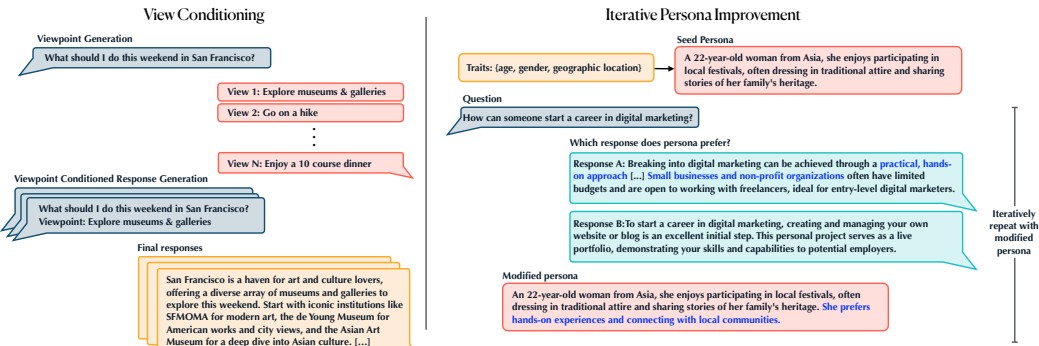

Figure 3: **Two key components in our synthetic data pipeline to aid with diversity and structure.** The left panel illustrates our method for increasing data diversity: we prompt a model to generate multiple viewpoints for a question and then condition our final response generation on these viewpoints. This yields greater diversity than temperature-based sampling. The right panel describes iterative persona improvement. If a seed persona is too underspecified for a clear preference, we iteratively refine its definition until it can make a robust prediction.

user's education background (**ELIX**), and personalizing for general question answering (**Roleplay**). We open-source preference datasets and evaluation protocols from each of these tasks for future work looking to study personalization (sample in supplementary).

**Reviews.** The Reviews task is inspired by the IMDB dataset [24], containing reviews for movies. We curate a list of popular media such as movies, TV shows, anime, and books for a language model to review. We consider two independent axes of variation for users: sentiment (positive and negative) and conciseness (concise and verbose). Here being able to pick up the user is crucial as the users from the same axes (e.g positive and negative) would have opposite preferences, making this *difficult* to learn with any population based RLHF method. We also study the steerability of the model considering the axes of verbosity and sentiment in tandem (e.g positive + verbose).

**ELIX.** The Explain Like I'm X (ELIX) task is inspired by the subreddit "Explain Like I'm 5" where users answer questions at a very basic level appropriate for a 5 year old. Here we study the ability of the model to personalize a pedagogical explanation to a user's education background. We construct two variants of the task. The first variant is **ELIX-easy** where users are one of 5 education levels (elementary school, middle school, high school, college, expert) and the goal of the task is to explain a question such as "How are beaches formed?" to a user of that education background. The second, more realistic variant is **ELIX-hard**, which consists of question answering at a high school to university level. Here, users may have different levels of expertise in different domains. For example, a PhD student in computer science may have a very different educational background from an undergraduate studying studying biology, allowing for preferences from diverse users (550 users).

**Roleplay.** The Roleplay task tackles general question answering across a wide set of users, following PRISM [19] and PERSONA Bench [5] to study personalization representative of the broad human population. We start by identifying three demographic traits (age, geographic location, and gender) that humans differ in that can lead to personalization. For each trait combination, we generate 30 personas, leading to 1,500 total personas. To more accurately model the distribution of questions, we split our questions into two categories: global and specific. Global questions are general where anyone may ask it, but specific questions revolve around a trait, for example an elderly person asking about retirement or a female asking about breast cancer screening.

One crucial detail for each task is the construction of a preference dataset that spans multiple users. But how should one construct such a dataset that is realistic and effective?

## 6 Sim2Real: Synthetic Preference Data Transfers to Real Users

Collecting personalized data at scale presents significant challenges, primarily due to the high cost and inherent unreliability of human annotation. Curating a diverse set of users to capture the full spectrum of real-world variability further complicates the process, often limiting the scope and representativeness of the data. Synthetically generating data using a language model [22, 1] is a promising alternative, since it can both reduce costly human data generation and annotation and

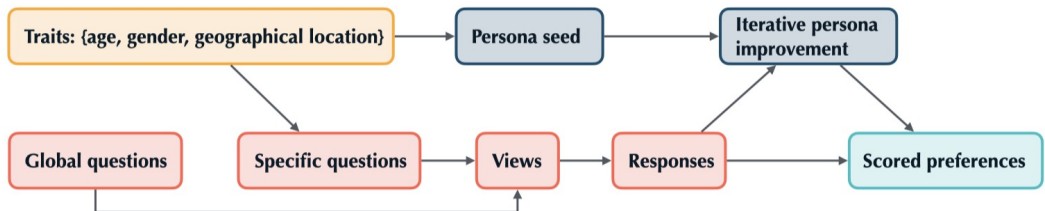

Figure 4: **Flowchart of Roleplay dataset generation:** Starting from a set of traits, a seed persona is constructed and a set of specific questions about that trait. Then responses are constructed with View-Conditioning. The seed personas are then iteratively refined to not be underspecified. Finally, the refined persona is used to score consistent preferences.

streamline the data curation process. We note that the use of synthetic data for personalization is nuanced and amenable in many applications, as explored in Appendix A.11. Can we generate diverse user preference data using language models in a way that transfers to real people?

We draw inspiration from simulation-to-real transfer in non-language domains like robotics [25] and self-driving cars [49], where the idea of domain randomization [44] has been particularly useful in enabling transfer to real environments. Domain randomization enables efficient adaptation to novel test scenarios by training models in numerous simulated environments with varied, randomized properties, enabling transfer to a held-out, real environment through interpolation.

But why is this relevant to personalization? As mentioned previously, each user can be viewed as a different "environment" to simulate as each user has a unique reward function that is represented by their preferences. To ensure models trained on synthetic data generalize to real human users, we employ domain randomization to simulate a diverse set of synthetic preferences. However, diversity alone isn't sufficient to learn a personalized LM. As studied in prior work [16, 50], it is crucial that the task distribution in meta-learning exhibits sufficient structure to rule out learning shortcuts that do not generalize. But how can we elicit both **diversity** and **structure** in our preference datasets?

**Encouraging diversity.** Diversity of data is crucial to learning a reward function that generalizes across prompts. Each domain has a slightly different generation setup as described in Section 5, but there are some general design decisions that are shared across all tasks to ensure diversity.

One source of diversity is in the questions used in the preferences. We use a variety of strategies to procure questions for the three tasks. For question selection for ELIX, we first sourced questions from human writers and then synthetically augmented the set of questions by prompting GPT-4o [31] with subsets of these human-generated questions. This allows us to scalably augment the human question dataset, while preserving the stylistic choices and beliefs of human writers. For the reviews dataset, we compiled a list of popular media from sites such as Goodreads, IMDb, and MyAnimeList. For the Roleplay dataset, we prompted GPT-4o to generate questions all users would ask (global) or questions only people with a specific trait would ask (specific). This allows us to have questions that are more consistent with the distribution of questions people may ask.

Additionally, having a diversity of responses is crucial for not only training the model on many viewpoints but also reward labeling, allowing for greater support over the set of possible responses for a question. To achieve diverse responses, we employ two strategies: Persona Steering [7] and view conditioning (Figure 3; left). For ELIX and Reviews, we use persona steering by prompting the model with a question and asking it to generate an answer for a randomly selected persona. For Roleplay, the user description was often underspecified so responses generated with persona steering were similar. Therefore, we considered a multi-turn approach to generating a response. First, we asked the model to generate different viewpoints that may be possible for a question. Then, conditioned on each viewpoint

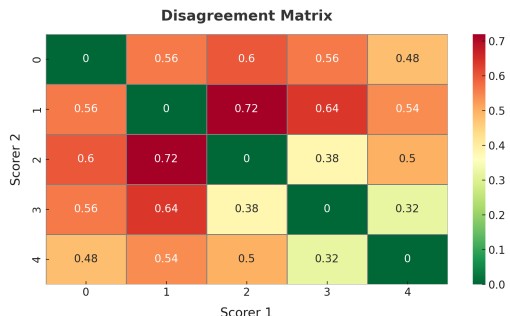

Figure 5: **Disagreement matrix across 5 users in Roleplay.** Here we plot the disagreement of preferences for 5 users. There is a mix of users with high and low disagreement.

independently, we prompted the model with the question and the viewpoint and asked it to answer the question adhering to the viewpoint presented. For example, if you consider the question, "How can I learn to cook a delicious meal?", one viewpoint here could be "watching a youtube video", better suited for a younger, more tech savvy individual, whereas viewpoints such as "using a recipe book" or "taking a cooking class" may be better for an older population or those who would have the time or money to spend on a cooking class. This allowed for more diversity in the responses and resulting preferences.

Finally, we sampled responses from an ensemble of models with a high temperature, including those larger than the base model we fine-tuned such as Llama 3.3 70b [14] and Gemma 2 27b [43], allowing for better instruction following abilities of the fine-tuned model, than the Llama 3.2 3B we fine-tune.

**Encouraging task structure.** Meta-learning leverages a shared latent structure across tasks to adapt to a new task quickly. The structure can be considered as similar feature representations, function families, or transition dynamics that the meta-learning algorithm can discover and leverage. For a preference dataset, this structure can be represented as the distribution of preferences across different users and is controlled by the scoring function and the distribution of responses.

One thing we controlled to enable better structure is the scoring function used to generate synthetic preferences. Firstly, we wanted to ensure consistent preference labeling. We use AI Feedback [1] to construct this, using relative pairwise feedback for preference labels, akin to AlpacaEval [11], as an alternative to absolute rubric based scoring, which we found to be noisy and inaccurate. The preference label along with being conditioned on the prompt, response, and general guidance on scoring, is now also conditioned on the scoring user description and additional scoring guidelines for user-aware preference labeling. Additionally, due to context length constraints, many responses for our preference dataset are shorter than the instruct model that we fine-tune from. Therefore, we prompt the model to ignore this bias. Furthermore, we provide each preference example to the model twice, flipping the order of the responses, and keeping filtering out responses that are not robust to order bias for both training and evaluation (win rates).

Additionally, as mentioned above, in some cases, such as with the Roleplay dataset, the user description is underspecified, leading to challenges in labeling consistent preferences. For example, if a user description does not have information about dietary preferences, inconsistency may arise for labeling preferences about that topic. For instance, in one preference pair, vegan cake recipes may be preferred but in another, steakhouses are preferred for date night. To fix this, we take an iterative process to constructing user descriptions. Firstly, we start with a seed set of user descriptions generated from the trait attributes. After generating questions and responses based on these seed descriptions, we take a set of question and response pairs. For each pair, we iteratively refine (Figure 3; right) the user description by prompting a model like GPT-4o to either label the preference pair or if the user description is insufficient, to randomly choose a preference and append information to the description so a future scorer would make the same decision. Finally, we utilize the updated user description to relabel preferences for the set of questions and responses allocated to that user with the labeling scheme above. This fix for underspecification also helps the COT prediction as predicting an underspecified user persona, can lead to ambiguous generated descriptions.

Finally, we desire structured relationships between users. To ensure this, we analyzed the disagreement (average difference of preference labels) of user's preferences across prompts to understand where users agreed and disagreed, and regenerated data if this disagreement was too high across users. By having users with some overlap, meta-learning algorithms can learn how to transfer knowledge effectively from one user to another. A sample disagreement plot for a subset of users in the Roleplay task can be found in Figure 5. We outline our full dataset generation process in Figure 4 in the Roleplay Task, starting from just a simple set of demographic traits.

| Strategy | Mean Similarity ($\downarrow$) | Median Similarity ($\downarrow$) |
|---|---|---|
| Llama 3.2 3B Instruct, temp. $= 0.3$ | 0.96 | 0.97 |
| Llama 3.2 3B Instruct, temp. $= 1.0$ | 0.94 | 0.95 |
| Llama 3.2 3B Instruct + persona steering (ours) | 0.81 | 0.82 |
| Llama 3.2 3B Instruct + view steering (ours) | 0.78 | 0.78 |
| Ensemble of Models + view steering (ours) | **0.71** | **0.73** |

Table 1: Comparison of diversity-inducing strategies as evaluated under ALOE [46].

| Method | Trained | Interpolated |
|---|---|---|
| Llama 3.2 3B Instruct | 50.0 | 50.0 |
| 4-shot Prompted | 66.6 | 61.9 |
| 4-shot Pref-FT | 66.5 | 66.1 |
| 4-shot FSPO (Ours) | **78.4** | **71.3** |
| 8-shot Prompted | 69.1 | 59.1 |
| 8-shot Pref-FT | 65.6 | 70.7 |
| 8-shot FSPO (Ours) | **80.4** | **73.6** |
| 8-shot FSPO + RAT (Ours) | **92.3** | **84.6** |

Table 2: Review Winrates

| Method | Winrate (%) |
|---|---|
| Llama 3.2 3B Instruct | 50.0 |
| IPO | 72.4 |
| Few-shot Prompting | 63.2 |
| Few-shot Pref-FT (GPO [54]) | 62.8 |
| RIC [48] | 53.3 |
| VPL [34] | 67.3 |
| FSPO (Ours, DPO) | **81.3** |
| FSPO (Ours, IPO) | **82.6** |
| FSPO + RAT (Ours, IPO) | **90.3** |
| Oracle (prompt w/ g.t. persona) | 90.9 |

Table 3: Winrates on Roleplay (1500 users)

| Method | ELIX-easy | ELIX-hard |
|---|---|---|
| Llama 3.2 3B Instruct | 50.0 | 50.0 |
| Few-shot Prompted | 92.4 | 81.4 |
| Few-shot Pref-FT | 91.2 | 82.9 |
| FSPO (Ours) | **97.8** | **91.8** |

Table 4: Winrates ELIX (550 users)

| Baseline Method | Winrate (%) |
|---|---|
| FSPO vs Base | **68.2** $\pm$ 1.93 |
| FSPO vs SFT | **72.3** $\pm$ 1.34 |

Table 5: Roleplay: Human Eval Winrates

**Evaluating diversity and structure.** We evaluate our design decisions with the following vignettes. For diversity, we measure semantic similarity using the dense score from the BGE-M3 model, following ALOE [46], on 100 randomly sampled prompts and 10 responses per prompt in the Roleplay task. As seen in Table 1, our proposed steering and ensembling mechanisms result in the base Llama 3.2 3B Instruct model exhibiting significantly reduced mean similarity. For structure, we estimate binary Shannon entropy of the preference label before and after iterative refinement. We condition on the persona and an unlabeled preference tuple (prompt and responses) and sample a preference label with a fixed temperature of 1.0 on 100 randomly sampled prompts from the Roleplay task with 100 pairs of personas and 10 samples per prompt. We use GPT-4o as the scoring model. Iterative persona refinement causes the entropy to drop from **0.64 nats to 0.13 nats**, validating the efficacy of this approach in inducing better persona-prompt-response consistency. For further validation, we show the efficacy of scaling the size of the dataset with respect to the amount of preference data and the number of few-shot examples in Table 7a and Table 7b, showing a monotonic increase in end-to-end performance. Furthermore qualitative examples in Appendix A.3, showcase the diversity of viewpoints and personas as well as their alignment when scoring for structure.

# 7 EXPERIMENTAL EVALUATION

**Baselines.** We compare FSPO against five baselines: (1) a base model generating user-agnostic responses, (2) few-shot prompting with a base model, following Meister et al. [26], (3) few-shot supervised fine-tuning (Pref-FT) based off the maximum likelihood objective from GPO [54], (4) prompting with an oracle user description following Persona Steering [7], and (5) Rewards-in-Context [48]. Specifically, for (1) we use a standard instruct model that is prompted solely with the query, resulting in unconditioned responses. For (2) and (3), the base instruct model is provided with the same few-shot personalization examples as in FSPO, but (2) zero-shot predicts the preferred response and (3) is optimized with SFT to increase the likelihood on the preferred response. In (4), the base model is prompted with the oracle, ground truth user description, representing an upper bound on FSPO's performance.

**Synthetic winrates.** We first generate automated win rates using the modified AlpacaEval procedure from Section 6. In the ELIX task in Table 4, we study two levels of difficulty (easy, hard), where we find a consistent improvement of FSPO over baselines. Next, in Table 2 for the Review task, on both Trained and Interpolated Users, FSPO allows for better performance on held-out questions. Finally, in Table 3, we study Roleplay, scaling to 1500 real users, seeing a win rate of 82.6% on both held-out users and questions. Also, RAT closes the gap to the oracle response, effectively recovering

the ground-truth user description. In Section A.2, sample generations from FSPO show effective personalization to the oracle user description. Given this result, can we personalize to real people?

**Preliminary human study.** We evaluate our model trained on the Roleplay task by personalizing responses for *real human participants*. We build a data collection app (Figure 7), interacting with a user in two stages. First, we ask participants to label preference pairs, used as the few-shot examples in FSPO. Then, for held out questions, we show a user a set of two responses: (1) a response from FSPO personalized based on their preferences and (2) a baseline response. Prolific is used to recruit a diverse set of study participants, evenly split across genders and continents, corresponding to the traits used to construct user descriptions. Question and response order is randomized to remove confounding factors. We evaluate with 50 users and 11 questions. As seen in Figure 5, we find that FSPO has a 68% win rate over the Base model and a 72% win rate over an SFT model trained on diverse viewpoints from the preference dataset. To assess statistical significance, we performed a one-sided binomial test. Here, the null hypothesis is that the probability of success is less than or equal to 50%, (ie, that our model is no better than the baseline) and the alternative hypothesis is that the probability is greater than 50%. The resulting p-value is 5.65e-09, so we reject the null hypothesis at any conventional significance level. We also validate FSPO on PRISM (Appendix A.10), a preference dataset on value based alignment from the community, showcasing benefits beyond our constructed datasets on real human users.

## 8 DISCUSSION AND CONCLUSION

We introduce FSPO, a novel framework for eliciting personalization in language models for open-ended question answering that models a distribution of reward functions to capture diverse human preferences. Our approach leverages meta-learning for rapid adaptation to each user, addressing limitations of conventional reward modeling techniques that learn from aggregated preferences. Through rigorous evaluation in 3 domains, we demonstrate that FSPO's generations are consistent with user context and preferred by real human users. Our findings also underscore the importance of diversity and structure in synthetic personalized preference datasets to bridge the Sim2Real gap. Overall, FSPO is a step towards developing more inclusive, user-centric language models.

## 9 ETHICS STATEMENT

While FSPO improves inclusivity by modeling diverse preferences, the risk of reinforcing user biases (echo chambers) or inadvertently amplifying harmful viewpoints requires careful scrutiny. Future work should explore mechanisms to balance personalization with ethical safeguards, ensuring that models remain aligned with fairness principles while respecting user individuality. Note, we choose to omit value-based personalization in the experiments as explored in works such as PRISM and Persona, instead focusing on the recommendation style of preferences such as travel preferences, where potential amplification of biases would be benign, having a limited effect on marginalizing particular subpopulations. Thus, this potential issue is a concern about using the algorithm in political or value-based contexts, not something that has arisen in the fine-tuned model. That being said, we do not explicitly mitigate this, which we leave to future work. Here, approaches such as Persona Vectors, recently released by Anthropic, can potentially be paired with an approach like FSPO to mitigate such biases in the training. We wish to emphasize clearly that our human study involves no collection of identifiable information and is strictly non-longitudinal, involving harmless, recommender-style questions. Under the criteria for Non-Medical IRBs, our study explicitly falls within the exemption specified by 45 CFR 46.104(d). Previous guidance received from our institutional IRB also confirms exemption status for such survey-based studies. Additionally, no such IRB was required in prior work including Direct Preference Optimization (DPO), AlpacaFarm, Chatbot Arena, and Persona, for nearly identical user study formulations. Thus, we strongly assert that formal IRB approval is unnecessary for our work. We additionally utilized LLMs such as GPT5/Gemini for minor rewritings of different sections throughout the paper for better readability.

## 10 REPRODUCABILITY STATEMENT

To ensure the reproducibility of our results, we provide a comprehensive account of our methodology, code, and data. The source code for our models and experiments is available in the supplementary materials and at the following anonymous repository: https://anonymous.4open.science/r/anon_fspo-E8FD/ (with dataset links anonymized and altered for final release). Our implementation is built upon the Pytorch FSDP framework, as utilized in the Direct Preference Optimization Codebase (https://github.com/eric-mitchell/direct-preference-optimization). All experimental details, including hyperparameter settings, are documented in Appendix A. The computational experiments were conducted on a machine with NVIDIA A100 GPUs and the required software dependencies are listed in the requirements.txt file within our code repository. The datasets used in our experiments will be made publicly available. We include samples of the dataset in the Appendix along with specific splits and any preprocessing steps applied to the data.

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

## A APPENDIX

### A.1 HYPERPARAMETERS FOR FSPO

| Name | Value |
|------|-------|
| Learning Rate (SFT/Pref-FT) | $1e^{-5}, 1e^{-6}, \mathbf{1e^{-7}}$ |
| Learning Rate (IPO) | $1e^{-5}, \mathbf{1e^{-6}}, 1e^{-7}$ |
| Beta (IPO) | $0.1, 0.05, 0.01, \mathbf{0.005}, 0.001$ |
| Number of Shots | $4, \mathbf{8}$ |
| Model Name | Llama 3.2 3B Instruct [14] |

Table 6: Sweep over hyperparameters for FSPO, recommended hyperparameters in bold.

#### A.1.1 ADDITIONAL ABLATIONS

We perform two ablations to study the impact of the size of the preference dataset and number of few-shot examples on performance. We see a monotonic increase in performance over the size and the number of fewshot examples in the Roleplay dataset.

| Preference Data (%) | Winrate (%) |
|---|---|
| 10 | 70.1 |
| 25 | 69.5 |
| 50 | 78.3 |
| 100 | 82.6 (reported) |

(a) Varying percentage of preference data.

| Few-Shot Examples | Winrate (%) |
|---|---|
| 1 | 65.7 |
| 2 | 69.3 |
| 4 | 72.1 |
| 8 | 82.6 (reported) |

(b) Varying number of few-shot examples.

Table 7: Ablation studies on roleplay task winrates with held-out synthetic users: (a) effect of preference data percentage per user, and (b) effect of number of few-shot examples.

### A.2 SAMPLE PERSONALIZED RESPONSES

We provide sample responses from FSPO in Figure 6 across the 3 tasks that were studied (ELIX, Reviews, and Roleplay). We additionally include the oracle scoring description for each response, demonstrating FSPO's ability to personalize. Additionally, for the Review task, we show responses for interpolated users which were not found in the preference dataset, showing the ability of FSPO to generalize and be controllable.

#### A.2.1 ROLEPLAY USER DESCRIPTION RATIONALIZATION (RAT)

```
QUESTION:
What are some good volunteer opportunities for me?

TRUE PERSONA:
A 53-year-old woman living in North America, an avid traveler and
    writer who shares her adventures through a popular travel blog
    . She loves spontaneous adventures and values long-term
    financial security. She is at a point in her career where
    opportunities for advancement might be time-sensitive. She
    enjoys movies that challenge perspectives and spark
    discussions. She prioritizes well-being over being fully
    informed. She is interested in the van life movement. She
    appreciates creativity and variety in meal options. She enjoys
     engaging in creative and tactile activities and prefers
    traditional learning methods. She values compassionate and
```

responsible choices. She enjoys exploring new and dynamic experiences. She is concerned about fast food. She values mental well−being and mindfulness. She enjoys attending events and conferences to meet new people. She appreciates practicality and health. She enjoys shows with dark humor and suspenseful storytelling.

RAT GENERATED PERSONA:
Based on the examples provided, the user appears to value a more holistic and experiential approach to learning and personal growth. They seem to prioritize human connection, nature, and personal development over technology and material wealth.

This user is likely someone who is interested in personal stories, emotions, and experiences, as evident from their appreciation for Sylvia Plath's poetry and the suggestion to weave storytelling into public speaking. They may also be someone who values authenticity, simplicity, and community, as seen in their suggestion of house swapping and volunteering as alternative vacation options.

In terms of lifestyle, this user likely prefers a balanced and flexible approach to life, allowing for a mix of work, personal growth, and leisure activities. They may also be someone who values environmental sustainability and social responsibility, as hinted at by their consideration of eco−friendly home improvements and sustainable financial planning. Overall, this user seems to be a thoughtful, empathetic, and nature−loving individual who prioritizes human connections, experiences, and personal growth over material possessions and technological advancements.

FINAL RESPONSE:
Based on your affinity for human connections, personal growth, and environmental sustainability, I'd like to suggest the following volunteer opportunities that might resonate with you:

**Community Garden or Urban Farm**: Many organizations in your area offer community gardens or urban farms where you can help cultivate and maintain a shared space, promoting sustainability and food security. This allows you to connect with nature and contribute to your community's well−being.

**Nature Reserves or Conservation Efforts**: Consider volunteering with local wildlife conservation organizations, such as bird watching tours, tree planting initiatives, or habitat restoration projects. This will enable you to work outdoors, connect with others who share your passion for nature, and contribute to preserving the environment.

**Community Centers or Environmental Education Programs**: Many community centers offer various programs, such as gardening workshops, environmental education classes, or sustainable living workshops. By volunteering at these centers, you can share your knowledge, learn from others, and make a positive impact on your community.

## Personalized Responses (ELIX-easy)

**Q: How does a thermostat control temperature?**

Oh boy, are you ready to learn about thermostats?
So, you know how sometimes your room gets too hot or too cold? That's because of something called temperature. Temperature is like a measure of how warm or cool things are. And thermostats are like superheroes that help control the temperature in your home.
Imagine you're playing with a thermostat in your room. It's like a special button that makes the temperature go up or down. But how does it actually work?

Thermostats are made up of three main parts:
1. Sensors: These are like special eyes that look at the room's temperature. They tell the thermostat how warm or cool it is.
2. Control Unit: This is like the brain of the thermostat. It looks at the sensor's readings and decides what to do next.
3. Heating/Cooling System: This is like the arm that actually moves. It heats or cools the room based on what the control unit tells it to do.

Here's how it works:
• The sensors send a message to the control unit, like "Hey, it's 75 degrees in here!"
• If the control unit thinks the room is too cold, it sends a signal to the heating system, saying "Turn on the heater!"
[...]

**Elementary School Student**

**Q: What makes diamonds so hard?**

Diamonds are renowned for their exceptional hardness, which is attributed to their unique crystal structure and chemical composition. Below is a breakdown of the factors contributing to diamond hardness:

**Key Factors Behind Diamond Hardness**

• Crystal Structure: Diamonds possess a face-centered cubic (FCC) crystal structure. Each carbon atom is bonded to four neighboring carbon atoms in a tetrahedral arrangement. This structure provides unparalleled strength and rigidity, enabling diamonds to resist deformation and fracture.
• Chemical Composition: Composed solely of carbon atoms arranged in a repeating pattern, diamonds owe their remarkable hardness to this carbon-rich composition.
• Tetravalent Bonds: Every carbon atom in a diamond is bonded to four neighbors through strong covalent bonds. These tetravalent bonds create a robust framework that supports the diamond's rigid crystal structure.

[...]

**Expert**

## Personalized Responses (Reviews)

**Write a review for the Action movie, The Dark Knight.**

The Dark Knight is a thrilling ride, but its flaws hold it back from being a truly great movie. The film's pacing is well-balanced, and the action scenes are intense and well-executed. The cast, led by Christian Bale and Heath Ledger, deliver solid performances. However, the plot is somewhat predictable, and the characters' motivations are not always clear. Overall, The Dark Knight is a fun, but forgettable, superhero film.

**Interpolated User: Concise + Negative**

**Write a review for the Action movie, The Dark Knight.**

*__A Flop of Catastrophic Proportions: "The Dark Knight Rises" Falls Woefully Short of its Pompous Promises__*

"The Dark Knight Rises," Christopher Nolan's vaunted swan song to the Batman franchise, is a calamitous, soulless, and creatively bankrupt exercise in superheroic excess, a monstrosity of monumental proportions that has left me questioning the very fabric of cinematic reality.

From its opening frames, it is evident that Nolan has traded his artistic integrity and attention to detail for a saccharine, crowd-pleasing formula that prioritizes style over substance, spectacle over sensibility. The narrative, a jumbled mess of convoluted plot twists, cheap thrills, and cringe-worthy romantic subplots, careens wildly from one paradox to the next, sabotaging any semblance of coherence or logical consistency.

Tom Hardy's Bane, the film's ostensibly formidable villain, is a paragon of cartoonish one-dimensionality, his motivations and demeanor reduced to a simplistic, mustache-twirling caricature of menace. The usually reliable Christian Bale, as Bruce Wayne/Batman, fares little better, his character reduced to a puzzle piece of angst-ridden introspection, his entire persona rendered inexplicably thin and unengaging.

[…] Continues

**Interpolated User: Verbose + Negative**

Figure 6: **Sample Personalized Response for ELIX (top) and Reviews (bottom).**

```
**Animal Shelters or Wildlife Rehabilitation Centers**: If you're
    an animal lover, consider volunteering at a local animal
    shelter or wildlife rehabilitation center. You'll have the
    opportunity to care for animals, interact with like−minded
    individuals, and contribute to animal welfare and conservation
    efforts.

**Neighborhood Events or Community Festivals**: Look for
    opportunities to volunteer at local events, such as festivals,
    farmers' markets, or street fairs. This allows you to engage
    with your community, promote sustainability, and build
    connections with neighbors through shared passion for
    environmental causes.

These opportunities will not only nurture your spirit of
    volunteering but also help you foster meaningful connections
    with nature, others, and your community while nurturing a
    sense of personal growth and fulfillment.
```

### A.3 Additional Details on Dataset Construction and Evaluation

**Dataset Details** We include a representative subset of the roleplay task and a jupyter notebook demonstrating shot construction for training and evaluation in the supplementary material (due to

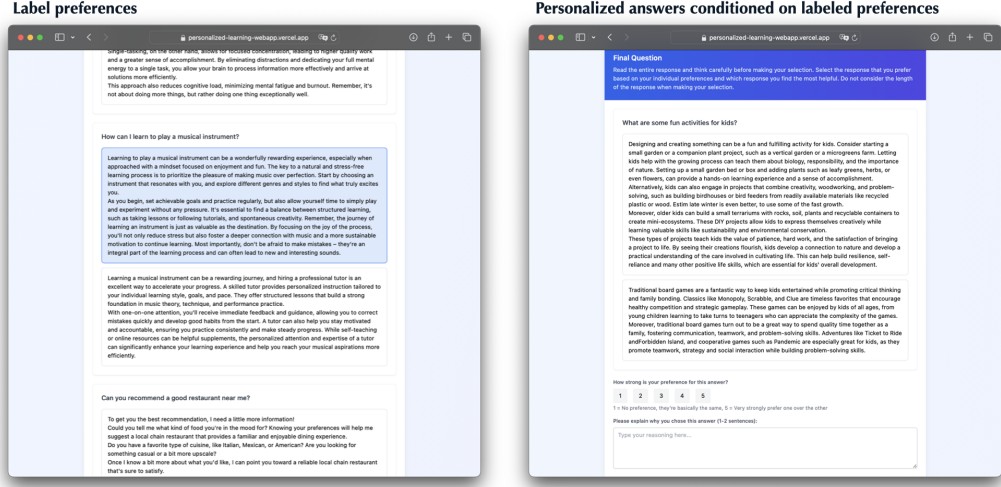

Figure 7: **An overview of the Human Study Interface.** First, users label a set of preferences. Then, a set of personalized answers are provided, conditioned on label preferences.

size restrictions). We will release the full datasets for each task in the final release of the paper. Below in Figure 8, we provide a qualitative example of a preference from the larger persona dataset, with details about the scoring persona and the viewpoints that underlie the responses. Additionally in Figure 10 and Figure 9, we showcase the qualitative diversity and alignment of viewpoints and personas present in the Roleplay task, enabling for preferences to have both diversity and structure.

| Key | Value |
| --- | --- |
| **Prompt** | Can you suggest a movie for family night? |
| **Scoring Persona** | A 46-year-old Asian female journalist, committed to uncovering the truth and reporting on important social issues affecting her community. She has a keen interest in learning new skills and sustainable practices and is open to unique travel experiences. She enjoys creative expression. She values making a positive impact. She appreciates variety and innovative experiences. She is interested in alternative philosophies and practices like Feng Shui. She prefers focusing on substance and content over presentation. She values meaningful and sincere connections over superficial ones. She is likely to appreciate diverse and practical solutions. |
| **Preferred View** | Suggest avoiding mainstream choices and opting for a lesser-known film that might spark interesting discussions. |
| **Preferred Response** | Instead of reaching for the usual Hollywood blockbuster, why not try "Hunt for the Wilderpeople"? This heartwarming indie film from New Zealand tells the story of a rebellious city kid who forms an unlikely bond with his foster uncle while on the run in the bush. It's funny, touching, and boasts stunning scenery, offering a great opportunity to discuss themes like found family, belonging, and escaping societal norms. It might even inspire some adventurous spirit in your family! You can find it streaming on several platforms. Give it a try – I think you'll be pleasantly surprised. |
| **Dispreferred View** | Go for an educational film like "The Secret Life of Trees," which can be entertaining and enlightening for everyone. |
| **Dispreferred Response** | For a unique and enriching family night, consider watching "The Secret Life of Trees." This documentary delves into the fascinating world of trees, exploring their communication networks, complex relationships, and vital role in our ecosystem. While seemingly focused on botany, the film weaves in captivating visuals and engaging storytelling that appeals to viewers of all ages. You'll learn about the surprising ways trees interact with each other, support wider ecosystems, and even impact human wellbeing. "The Secret Life of Trees" offers a powerful blend of education and entertainment, sparking conversation and fostering a deeper appreciation for the natural world. |

Figure 8: **Sample Preference (Roleplay).** In this preference example, we provide a sample prompt, scoring persona, (dis)preferred view, and (dis)preferred response.

**Evaluation Prompt for Synthetic Preferences**  We used GPT-4o as a Judge using a modified variant of the Alpaca Eval [11, 10] Prompt to be aware of a user description when scoring preference examples.

Here is the system prompt:

```
You are a highly efficient assistant, who evaluates and selects
    the best large language model (LLMs) based on the quality of
    their responses to a given instruction. This process will be
    used to create a leaderboard reflecting the most accurate and
    human-preferred answers.
```

Here is the user prompt:

| Question | Viewpoint 1 | Viewpoint 2 | Viewpoint 3 | Viewpoint 4 | Viewpoint 5 |
|---|---|---|---|---|---|
| Can you suggest some popular anime to watch? | A newcomer seeking an easy entry point | A genre enthusiast (e.g. mecha, slice-of-life, shōnen) | A parent concerned about age-appropriate content | A critic scouting award-winning or groundbreaking series | A nostalgic fan revisiting classics |
| How can I travel around Europe on a budget? | A student backpacker aiming to minimize accommodation and transport costs | A digital nomad balancing work commitments and exploration | A family looking for affordable, kid-friendly activities | An eco-conscious traveler preferring slow travel and local stays | A luxury traveler hunting budget hacks without sacrificing comfort |
| Can you recommend a healthy breakfast recipe? | A nutritionist focusing on optimal macro- and micronutrient balance | A busy professional needing something quick to prep | A vegetarian or vegan with specific dietary restrictions | A fitness enthusiast prioritizing high-protein options | A home cook looking to explore new, flavorful ingredients |
| What is the best way to handle workplace conflict? | A manager aiming to maintain team morale and productivity | An HR professional ensuring compliance with policy | An individual contributor seeking to assert personal boundaries | A mediator focusing on active listening and empathy | A cultural consultant navigating diverse communication styles |
| What is the most effective way to study for exams? | A student practicing spaced repetition and active recall | A visual learner creating mind-maps and diagrams | A collaborative learner using group study sessions | A planner using time-blocking and structured schedules | A tech-savvy learner leveraging apps, flashcards, and analytics |

Figure 9: **Sample Viewpoints (Roleplay).** For a given question, a diversity of viewpoints for a response can be inferred to create a preference dataset that encompasses a wide-range of opinions.

| Description | Age | Gender | Geographic Location |
|---|---|---|---|
| A 51-year-old Asian female artist, known for her vibrant paintings that reflect her cultural heritage and personal journey, often exhibited in galleries around the world. She is interested in stable long-term investments, appreciates films that tackle social issues, and prefers cost-effective travel solutions. She enjoys preparing simple and natural meals. She values self-reflection and personal growth. She enjoys connecting with nature and is interested in traditional medicine practices. She prioritizes ethical and responsible choices, such as adopting animals. She appreciates established and proven experiences. She prefers engaging in discussions and sharing expertise in online forums. She enjoys incorporating natural elements into her surroundings. She values cultural significance and community connections. She values character development and thought-provoking narratives in stories. | 51 | Female | Asia |
| A 56-year-old Swiss man who is a financial analyst, offering his expertise to international firms while enjoying the tranquility of the Swiss countryside. He has a keen interest in sustainable practices, local cuisine, and films with social commentary. He enjoys reading and discussing literature. He is budget-conscious and values cost-effective options, and appreciates character customization. He prefers practical solutions. He is particularly concerned with the health implications of cooking methods. He is familiar with technology and social media tools and values efficient and effective solutions. He is meticulous in research and decision-making. He prefers more thoughtful and natural approaches to personal care. | 56 | Male | Switzerland |
| A 23-year-old male from Africa who is an environmental activist, leading campaigns to raise awareness about climate change and its impact on local communities. He enjoys writing and is highly motivated and values financial independence to support his activism. He seeks fulfillment in both personal and professional areas. He enjoys exploring diverse culinary flavors and incorporating healthy plant-based meals into his lifestyle. He is interested in science and educational activities that relate to the environment. He values flexibility and adaptability. He prefers online interactions. He values understanding and socialization in animals. He appreciates natural light and its positive effects on mood and productivity. He values supporting local businesses and community sustainability. He enjoys dark humor. | 23 | Male | Africa |

Figure 10: **Sample Personas (Roleplay).** A comprehensive description of the user is iteratively refined from preference pairs for that user, seeded with attributes of age, gender, and geographic location.

```
You are tasked with evaluating the outputs of multiple large
    language models to determine which model produces the best
    response from a human perspective.

## Instructions

You will receive:
1. A **User Instruction**: This is the query or task provided to
    the models.
2. **Model Outputs**: Unordered responses from different models,
    each identified by a unique model identifier.
3. A **User Description**: This describes the user's preferences
    or additional context to guide your evaluation.

Your task is to:
1. Evaluate the outputs based on quality and relevance to the user
    's instruction and description.
2. Select the best output that meets the user's needs.

## Input Format

### User Instruction
{QUESTION}

### Model Outputs
- Model "m": {RESPONSE_A}
- Model "M": {RESPONSE_B}

### User Description
{USER_DESCRIPTION}

## Task

From the provided outputs, determine which model produces the best
    response. Output only the model identifier of the best
    response (either 'm' or 'M') with no additional text, quotes,
    spaces, or new lines.

## Best Model Identifier
```

**Additional Human Study Details**    As shown in Alpaca Eval 2.0 [10], several biases can affect the evaluation of language models such as length, format, and more. For this reason, we took action to normalize both FSPO and baselines in 3 different categories. First, length is an evaluation bias. For this reason, we computed the average length of responses from FSPO and prompted the base model during evaluation to keep its responses around the average length in words ($\approx 250$ words). For the SFT baseline, we found that this was consistent with FSPO since it was fine-tuned on the same preference dataset. Additionally, due to context length restrictions and the instruction following abilities of smaller open-source LLMs, we decided to have formatting be consistent as paragraphs rather than markdown for the Roleplay task. Thus, we similarly prompted the Base model with this behavior. Finally, a differing number of views can also skew the evaluation, as a large proportion of users seem to prefer direct answers. Additionally, if more views are presented, a user may prefer just one of the many views provided, skewing evaluation. Thus, we ensure that when two responses are compared, they have the same number of views. In future, work, it would be interesting to consider how to relax some of the design decisions needed for the human study. We additionally provide screenshots of the human study interface in Figure 7.

Below is the full text of instructions given to the participants:
"This is a study about personalization. You will be asked to read a set of 20 questions (9 on the first

page, 11 on the second page). For each question, there are two responses. Please select the response that you prefer. Make this selection based on your individual preferences and which response you find the most helpful. Read the entire response and think carefully before making your selection."

We utilize the demographic information that Prolific provides for each user such as their age group, continent and gender to chose questions but do not store that information about the user. We collect no identifying information about the user and will not make any of the individual preferences from a user public. We pay each user a fair wage subject to the current region that we reside in. We received consent from the people whose data we are using and curating as the very first question in our survey. The demographic and geographic characteristics of the annotator population is exactly the same as Prolific. We do no filtering of this at all.

### A.4 Training Details and Hyperparameters for FSPO and baselines

Similar to DPO [36] and IPO [12], we trained FSPO in a two stage manner. The first stage is Fewshot Pref-FT, increasing the likelihood of the preferred response. The second stage is Fewshot IPO, initialized from the checkpoint of Fewshot Pref-FT. One epoch of the dataset was performed for each stage. For the IPO baseline, we followed a similar procedure. Additional hyperparameters can be found in Table 6.

### A.5 Additional Details of Setup for Reproducability

We used both code, models, and data as scientific artifacts. In particular, for code, we built off of the codebase from Rafailov et al. [36], with an Apache 2.0 license. We additionally adapted our evaluation script from Alpaca EVAL, including the prompt, and other criterion for evaluation and normalization. We have reported the implementation details for synthetic evaluation in Section 6 and human study evaluation in Section A.3.

For models, we used a combination of open-source and closed-source models. The models that we used for sampling data are the Llama family of models [14] (Llama 3.2 3b, Llama 3.1 8b, Llama 3.3 70b) with the llama license (3.1, 3.2, 3.3), the Qwen family of models [35] (Qwen 2.5 3b, Qwen 2.5 32b, Qwen 2.5 72b) with the qwen license, the Gemma 2 family of models [43] (Gemma 2 2b, Gemma 2 9b, and Gemma 2 27b) with the gemma license, and the OpenAI [31] family of models (GPT4o, GPT4o-mini) with the OpenAI API License (based off of the MIT License). We used SGLang [55] and VLLM [20] for model inference. For training, we used 1 node of A100 GPUs (8 GPUs) for 8 hours for each experiment with FSDP. Cumulatively, we used approximately 4000 hours of GPU hours for ablations over dataset, architecture design and other details.

With respect to the dataset, for questions for the review dataset, we sourced media names from IMDb [17], Goodreads [13], and MyAnimeList [30]. We define the domains in more detail in section 5. Seed questions for ELIX were human generated, sourced from Prolific. The dataset is entirely in English, with some artifacts of Chinese from the Qwen model family, which will be filtered out for the final release of the dataset. None of this data has identifying information about individual people or offensive content as the dataset was sourced from instruction and safety-tuned models, with each step of the dataset having a manual check of the inputs and outputs. In terms of statistics of the dataset, the review dataset has 130K train/dev examples and 32.4K test examples, the ELIX-easy dataset has 235K train/dev examples and 26.1K test examples, the ELIX-hard dataset has 267K train/dev examples and 267K test examples, and the roleplay dataset has 362K train/dev examples and 58.2K test examples, with a total of 1.378 million examples. For our statistics, we reported the average winrate % for each method on both synthetic and human evals, following prior work in alignment like AlpacaFarm [11].

Each of the artifacts above was consistent with its intended use and the code, models, and datasets should be usable outside of research contexts.

### A.6 Synthetic Data is NOT limited by what is internalized by the LLM

Though the seed persona is instantiated and refined with an LLM, one part of the refinement strategy that potentially mitigates the stereotype concern that you have raised is that we randomly select a response to be preferred from a choice of two viewpoint-conditioned responses to augment the seed persona. Therefore, through the refinement process, we recover a persona description that could map to any permutation of the $2^N$ preferences, allowing for more expressivity than what is internalized by the LLM. Additionally, in the viewpoint generation process, we ask the model to list

multiple viewpoints for a particular question, which allows the model to elicit a diverse set of possible responses to score and iteratively refine the persona with. This additionally reduces the occurrence of "stereotypical personas", allowing for more nuanced answers for a particular question. In Figure 9, we list 3 sample personas to qualitatively show their diverse nature.

## A.7 SAMPLING OF PREFERENCES PER USER

As seen in Algorithm 1, line 5, for each user, we sample a subset of the user's preferences to construct the few-shot preferences for that user during training. During training, we revisit the user and resample a new subset of preferences. In Table 7, we show an ablation over the number of few-shot preferences that are sampled, and do see gains with the number of preferences conditioned on. For our synthetic evaluation, we match this form of sampling, drawing multiple sets of few-shot preferences per held-out user and averaging over the set to construct the win-rate per user, which we further aggregate over all users. For the human study, due to cost constraints, we ask participants to label a fixed set of preference pairs in the first stage of our study, used as the few-shot examples. Then, for several held-out questions, we evaluate for this fixed set a response from FSPO and a baseline model.

This training and evaluation procedure mitigates the concern that the choice of the N few shot examples impacts performance.

## A.8 ADDRESSING THE ADDITIONAL OVERHEAD OF FEW-SHOT PERSONALIZATION

Few-shot preferences do expand the context requirements of an LLM. One approach to mitigate this is the RAT prediction, which can be inferred from the user's preferences and may be shorter than the preferences themselves to condition on. Furthermore, this can be cached for a user to mitigate latency issues and used across different prompts. Finally, models today are continuing to scale the length of their context (such as Gemini 2.5 pro having over a 1 million tokens in context) so this may be a small price to pay with respect to the overall context.

## A.9 LIMITATIONS

Our human study was preliminary with control over the questions that a user may ask, format normalization where formatting details such as markdown are removed, and view normalization comparing the same number of viewpoints for both FSPO and the baselines. To the best of our knowledge, we are the first to perform such a human study for personalization to open-ended question answering. Future work should do further ablations with human evaluation for personalization. Additionally, due to compute constraints, we work with models in the parameter range of 3B (specifically Llama 3.2 Instruct 3B) with a limited context window of 128K, and without context optimization such as sequence parallelism [23, 47], further limiting the effective context window. It is an open question on how fine-tuning base models with better long-context and reasoning capabilities would help with FSPO for personalization, such as the 2M context window of Gemini Flash Thinking models, especially in the case of RAT.

## A.10 FSPO ON HUMAN PREFERENCE DATASET (PRISM ALIGNMENT)

We have run FSPO on the PRISM Alignment Dataset. For evaluation, we evaluate FSPO as a reward model (leveraging the duality of DPO and IPO) by comparing the log likelihood of the preferred response and dispreferred response on held-out preferences. On this dataset, we achieve a reward prediction accuracy of 82.8%, whereas population based approaches such as IPO achieve a reward prediction accuracy of 61.7%, showcasing the efficacy of the method in generalizing to a held-out user. There is no protocol for evaluating generated responses on PRISM, as the survey provided per user is highly underspecified, providing little to no details about the user for response evaluation.

## A.11 ADDITIONAL DISCUSSION OF THE USE OF SYNTHETIC DATA

It would be ideal to use a large-scale real user preference dataset suitable for developing and testing robust personalization systems. Unfortunately, in the open-source community, no such high-quality dataset exists, necessitating the generation of a synthetic preference dataset. In the related work, we do consider a prior human collected dataset, the Prism Alignment Dataset [3], where we find that a proportion of the prompts are of lower quality (such as including conspiracies such as "i think the moon landing was faked") and quite distinct from questions that a user would ask an assistant, focusing on value-based personalization (such as "Who is right in the Hamas-Israeli war? Hamas or the Israelis?"), which have troubling ethical considerations. In contrast, the Roleplay synthetic dataset studies more natural, recommendation style questions that involve personalization such as "What

should I do this weekend in San Francisco?" or "Can you recommend a good podcast?", synthetically augmented from seed human generated questions.

Thus, the synthetic data pipeline from FSPO can be a practical solution for scenarios where high-quality, task-specific preference data is unavailable, sparse, or lacks diversity. In these situations, our approach can supplement and augment existing real data, rather than merely replacing it. Below, we will describe some real-world problem instances where FSPO can be beneficial.

**1. Cold Start Problem** One advantage of the synthetic construction proposed is addressing the cold-start problem. When launching a new personalized feature, there is often no historical data to draw upon. FSPO provides a robust initial data curation pipeline that can deliver immediate value, as evidenced in the tasks studied in this work. This extends to situations where an organization has a wealth of user data, but not in a format amenable to LLMs (e.g, a housing and neighborhood commerce network such as BILT, which has a set of user transaction patterns and platform engagement not standard to LLMs). In such instances, a synthetic preference dataset can be designed using FSPO, based on the existing data signals. Additionally, as real preference data is collected, it can be integrated with or used to fine-tune the synthetically trained model, demonstrating how FSPO can serve as a critical foundation and accelerator. Furthermore, works such as AlpacaFarm have been introduced for prototyping/development of preference-based systems. As stated in their abstract, synthetic data such as LLM Prompts can simulate human feedback that is 50x cheaper than crowdworkers and display high agreement with humans (corroborated with our human study). Thus, in many real-world applications, this synthetic data generation pipeline can be used for benchmarking purposes that emulate a more realistic downstream application in personalization.

**2. Privacy-Sensitive Settings** Additionally, there exist applications where collecting and storing extensive user data is either impractical or undesirable due to privacy concerns. Consider an on-device AI assistant, a confidential workplace tool, financial/banking assistants, or a medical assistant. Here, approaches from the synthetic data pipeline, such as iterative persona construction (Figure 3, right) can be an appropriate approach to synthetically generate a user profile from the user preferences to elicit personalization, without needing a persistent user-written profile. This can additionally be constrained/controlled to not include any personally identifiable information from the preferences that are collected, which is advantageous, for example, in medical domains to avoid infringing on HIPAA. Collecting a comprehensive, detailed user profile is often intractable and inadvisable in such applications, but is beneficial for fine-tuning a personalized model, which our approach provides a controllable solution for. Similarly responses from a user may be difficult to collect in this instantiation as well. Here, our diverse response generation strategy may be a good fit, such as viewpoint conditioned responses, where viewpoints can be supervised by experts in a domain like medical professionals.

**3. Guided Data Curation & Metrics** Finally, our synthetic data pipeline is instantiated on the guiding principles of structure and diversity, theoretically motivated by task-generation in meta-learning, which are readily transferable to real tasks. To ground this in a real problem, let's consider the education domain that you have suggested, where student data might be available. Our approach can provide guidance on data curation or data selection for a personalized system in this domain. To concretely measure these principles, we study and empirically evaluate metrics that characterize the principle. For example, for diversity, we study the embedding similarity of responses as seen in Table 1, and we introduce a disagreement metric as seen in Figure 5 to capture the diversity of responses and users. This can be readily used to gauge the diversity of real preference data, such as capturing the diversity of education backgrounds of students or the diversity in tutoring conversations in an education setting. We characterize structure in preferences by the binary Shannon entropy of the preference labels, a metric we study in section 6. This can be used to identify underspecification of a user's education background with the prompts and responses that they label preferences for, and potentially be used to filter users in the dataset that may be too noisy. Having inconsistencies in preferences or the user description makes the learning problem much more difficult, as described in prior work such as C-DPO and IPO. Overall, these metrics can guide the data selection/curation process of existing human data in domains such as education and has been validated on users in our human study, which indicates the effectiveness of the data curation approach.

To sum up, FSPO can be readily incorporated into several real world applications, where it can help provide a warm-start in data limited regimes or a dataset to prototype with, with settings where user-

privacy is paramount, and additionally guide the curation and selection of human data, in applications such as education through the guiding principles of structure and diversity with the metrics proposed.

### A.12 THE USE OF LARGE LANGUAGE MODELS (LLMs)

Large Language Models are used to assist with proofreading and minor wording improvements. All research ideas, experiments, and conclusions were conceived and validated by the authors. Additionally, tools such as Cursor were utilized as coding assistants during the development of the coding infrastructure for the project.

