# OpenReview forum: "FSPO: Few-Shot Optimization of Synthetic Preferences Effectively Personalizes to Real Users"
_ICLR.cc/2026/Conference — ICLR 2026 Poster_

### Official Review · Reviewer_ehDx · 2025-11-01

**Soundness:** 3
**Presentation:** 2
**Contribution:** 3
**Rating:** 6
**Confidence:** 2

**Summary:**

The paper proposes Few-Shot Preference Optimization (FSPO) for fast, user-specific alignment from a small number of preference examples. It utilizes the idea of meta learning by viewing each user as a task and adapts the model by conditioning on the user’s few-shot preferences. The method also uses user description rationalization to infer a short natural-language profile from those examples. Experiments show consistent gains over non-personalized and standard preference-tuned baselines. Overall, the paper offers a clear formulation of few-shot personalization with a simple, practical recipe and strong results.

**Strengths:**

This is a solid paper. The FSPO approach is novel, and the problem, effective user-specific preference adaptation, is crucial for LLMs. Empirically, the gains are consistent across settings and baselines, with good sample efficiency on small backbones and few shots. Overall, the writing and ablations are clear and support the main claims.

**Weaknesses:**

1. In the experiment, the method adds test-time steps (especially with RAT), but there are no latency/throughput numbers—only a note that RAT or caching could mitigate context overhead.
2. Minor weaknesses in experiments. Heavy reliance on LLM-judged win rate; PRISM “transfer” is narrow in the Appendix; only the single 3B backbone is considered—no size scaling.
3. In equation (5), I feel the notation is ambiguous. For the historical data, an additional superscript may solve the potential misunderstanding.

**Questions:**

1. As noted in the weaknesses, what is the inference-time computation cost of RAT?
2. The experimental design and results are strong. However, why is the standard pairwise response selection task not included? There are many ready datasets and more baselines to compare.

---

> ### Author Response · Authors · 2025-11-17
> **Response to Reviewer ehDx [1/N]**
>
> We thank the reviewer for their insightful comments and feedback, particularly for acknowledging the novelty of our framework for user-specific preference adaptation and the sample efficiency of our approach.
>
> > Inference Time Cost of RAT
>
> For the roleplay task, the average RAT length was 239 tokens, while the corresponding response averaged 896 tokens. We argue that this additional computation is not a meaningful bottleneck for contemporary models, for two main reasons.
> First, modern reasoning-focused models—such as DeepSeek R1, GPT-5, and Gemini—are designed with extensive output and context token budgets, often exceeding 256k tokens, with Gemini 1.5 supporting over one million tokens of input context. These expanded capacities enable a more favorable trade-off between efficiency and accuracy, allowing practitioners to adjust for performance as needed without incurring prohibitive overhead.
>
> Second, the incremental cost of these additional context tokens can be largely amortized through prefix caching, a technique now standard in frontier and open-source inference engines alike (e.g., [OpenAI Prompt Caching](https://platform.openai.com/docs/guides/prompt-caching), [vLLM Automatic Prefix Caching](https://docs.vllm.ai/en/latest/features/automatic_prefix_caching/)). By reusing the key–value cache from prior interactions, subsequent queries can leverage precomputed context representations. This results in negligible increases in latency or memory footprint at inference time, even when RAT adds additional inference steps.
>
> > On Evaluation Method with LLMs + Human Study + Prism
>
> Synthetic preferences have been widely adopted as a substitute for human preference data in prior work, such as Constitutional AI [1] and RLAIF [2], and our approach should not be uniquely penalized for employing this as one of our evaluation methods. Nonetheless, we strengthen our evaluation with a real human study presented in Table 4, which demonstrates a statistically significant win rate of approximately 70%. This result indicates that our framework effectively transfers to real human judgments in response generation, extending beyond synthetic evaluation metrics.
>
> Furthermore, PRISM is a well-established dataset comprising sociodemographic information and stated preferences from 1,500 diverse human participants across 75 countries. While the dataset offers valuable coverage, it lacks a built-in evaluation protocol for response generation. As a result, we rely on response selection (reward accuracy) as a proxy metric.
>
> References
> 1. Constitutional AI: Harmlessness from AI Feedback
> 2. RLAIF vs. RLHF: Scaling Reinforcement Learning from Human Feedback with AI Feedback
>
> > Size Scaling
>
> Your points regarding the experimental scope of model size are well taken. Within our academic compute constraints, our primary goal was to validate the central hypothesis: that large language models can be effectively personalized to individual users by casting reward modeling as a meta-learning problem. This approach leverages diverse, structured synthetic data to rapidly infer personalized reward functions that generalize to real human users. Scaling to substantially larger models (e.g., 32B parameters) was infeasible given our budget and the long-context nature of our few-shot examples. Conversely, scaling down to smaller models would be ill-posed, as their limited context capacity would undermine the very capabilities our framework depends on.
>
> > Equation (5) Notation
>
> We apologize for the confusion. We now include a superscript to denote the historical dataset in the manuscript. We repeat it below for your reference. Please let us know if you have any further suggestions:
>
> ```
> \begin{align}
>     \min_{\theta} \mathbb{E}_{\mathcal{S}^{(i)} \sim \mathcal{S}} \left[\mathbb{E}_{(x,y_1,y_2,c) \sim \mathcal{D}_i, \{(x,y_1,y_2,c)\}_{1}^{N} \sim \mathcal{D}_i^{\textit{hist}}}\left[\mathcal{L}^{\theta}_{\textit{pref}}\left(x,y_1,y_2,c \vert \{(x,y_1,y_2,c)\}_{1}^{N} \right)\right]\right]
>     \label{eq:metalearn-obj}
> \end{align}
> ```

---

> > ### Author Response · Authors · 2025-11-17
> > **Response to Reviewer ehDx [2/N]**
> >
> > > Pairwise response selection task
> >
> > FSPO targets open-ended generation tasks for LLMs, in contrast to classification or structured-output tasks (such as pairwise response selection or survey-based approaches), which represent a much narrower form of personalization. This open-ended setting provides a more realistic avenue for personalization, enabling rich and diverse user preferences to manifest across full model responses rather than within a fixed set of predefined options. This direction aligns with current efforts by major LLM providers—such as OpenAI’s GPT-5.1 [1], which emphasizes response steering to make models feel uniquely tailored to individual users. Additionally, note that our evaluation on Prism in Appendix A.10 can be viewed as response selection: we leverage the duality between FSPO’s generation and reward modeling to assess reward model accuracy—specifically, how often the preferred response for a given user receives a higher reward than the dispreferred one. Here, we do find a benefit of FSPO with respect to baselines, with a reward prediction accuracy of 82.8%.
> >
> > **Please let us know if you find your concerns addressed, and if so, we would be grateful if you are willing to raise your score.**
> >
> > References
> > 1. GPT-5.1: A smarter, more conversational ChatGPT

---

> > > ### Author Response · Authors · 2025-11-26
> > >
> > > Dear Reviewer,
> > >
> > > Since there is only 1 week left in the rebuttal period, it would be helpful for us to know if our rebuttal above addressed all your questions and concerns, or if any others remain. We are happy to discuss further and provide more evidence if it is helpful. If all your concerns are addressed, we would be grateful if you could acknowledge that soon.
> > >
> > > Thanks,
> > > Authors

---

### Official Review · Reviewer_5VkS · 2025-11-02

**Soundness:** 3
**Presentation:** 3
**Contribution:** 2
**Rating:** 4
**Confidence:** 3

**Summary:**

Few-Shot Preference Optimization (FSPO) treat personalization as a black-box meta-learning problem where each user = a task. Given a small set of that user’s preference pairs (preferred vs dispreferred responses), the model must answer a new query in that user’s style. Instead of classic RLHF that averages over users and washes out individuality, FSPO keeps user IDs in the preference data and optimizes a DPO/IPO-style objective conditioned on that user’s few-shot prefs.

A key ingredient is RAT (User Description Rationalization): first, from the few-shot preferences, the model generates a natural-language user description (a synthetic persona); second, it generates the final response conditioned on that description. This makes the user representation more explicit and interpretable, and it’s like spending extra inference compute to do better reward modeling.

Because real personalized data is expensive, the paper builds a structured synthetic data pipeline: generate multiple viewpoints per question, refine personas iteratively when they’re underspecified, and enforce both diversity and structure so the model can transfer to real users (sim2real). They evaluate on three personalization-style tasks (Reviews, ELIX, Roleplay) and show that RAT + FSPO outperforms simpler few-shot prompting and standard preference learning, and argue this pipeline also helps cold start and privacy-sensitive settings.

**Strengths:**

Thoughtful synthetic data design (diversity + structure). The pipeline doesn’t just random-sample dialogues; it explicitly generates multiple viewpoints and refines personas to avoid underspecification.
Casting “each user = a task, few-shot prefs = task context” is a neat, well-motivated transfer of meta-learning ideas to RLHF-style preference learning. It directly addresses the criticism that population-level preference aggregation destroys individual preferences.

**Weaknesses:**

Sim2real is argued well, but the paper could show more quantified transfer: e.g. how much do metrics drop when you remove diversity, or structure, or persona refinement? Right now, some of the pipeline choices feel plausible but not fully ablated.
Judge / metric bias. Since the setup is preference-style, are judges from the same model family? Are they sensitive to verbosity / style? We don’t see a deep cross-judge analysis.

**Questions:**

RAT sensitivity. How sensitive is performance to incorrect or partially correct user descriptions? Can you show a curve where you corrupt 0%, 25%, 50% of the RAT description and see task win-rate drop?

---

> ### Author Response · Authors · 2025-11-17
> **Response to Reviewer 5VkS [1/N]**
>
> We thank the reviewer for their insightful comments and feedback, particularly for acknowledging the thoughtfulness of the synthetic data pipeline and our application of meta-learning ideas to RLHF preference learning.
>
> > Sim2real is argued well, but the paper could show more quantified transfer
>
> We emphasize that our evaluation assesses the system’s effectiveness fully end-to-end, achieving approximately a 70% win rate over baselines in a preliminary human study, with statistical significance testing confirming the validity of the results. This demonstrates that the synthetic data, as a whole, enables effective transfer to real human users.
>
> As an additional control, we now include an ablation removing View Conditioning and Iterative Persona Improvement, in which seed personas (as shown in Figure 3) are used both to generate responses and to judge them for preference data construction. This configuration yields a 56% win rate for FSPO over the SFT model—a significant drop in performance—highlighting the importance of structure and diversity in the preference data process. We will incorporate this result into the appendix of the work.
>
> Furthermore, we present quantitative analyses of our method’s components. Table 1 compares diversity-inducing strategies: ensembling plus view steering reduces the cosine similarity between generated responses. We also assess persona–prompt–response consistency using the sampling procedure in Section 6, showing that iterative persona refinement lowers entropy of sampled preferences from 0.64 to 0.13 nats. This quantitatively justifies the claims of structure and diversity in the synthetic generation process.
>
> Qualitatively, both components are essential: (1) structured persona evaluation prevents underspecified settings where a persona cannot reliably express a preference, and (2) diverse viewpoints enhance the model’s response diversity for individual users.
>
> > RAT sensitivity.
>
> In FSPO, user description rationalization (RAT) is produced from a user’s few-shot preference pairs and then used together with those very pairs to condition generation. Furthermore, RAT is trained via pairwise preference optimization such that, given the same few-shot preferences, descriptions that are semantically closer to the gold persona are preferred; generation then conditions on the prompt, the few-shot preferences, and the RAT description (two-stage pipeline in Fig. 2). This encourages the generated rationalizations to remain closely aligned with the true user profile. That said, quantifying the model’s sensitivity to corruption in the rationalized description is potentially ill-posed, especially when defining and applying “corruption” systematically, while avoiding distributional shift poses challenges. For instance, masking tokens may yield underspecified or ambiguous descriptions, primarily probing robustness to missing information rather than misinformation. Conversely, replacing tokens with random or contradictory content risks creating out-of-distribution inputs, where corrupted personas deviate significantly from those observed during training.
>
> Furthermore, we do not claim that the generated rationalizations are factually accurate. Rather, as with the chain-of-thought process in reasoning tasks, intermediate rationalizations for unseen users may contain minor inconsistencies relative to the true user description. However, the use of this additional inference computation provides a benefit at test time, generalizing responses to unseen users. Thus, the corruption experiment may be potentially ill-posed as we wish to test directly the model’s own ability to generate useful inference tokens that benefit both reward modeling and response generation, following works such as GenRM[1]. Empirically, we validate this claim, where on the Roleplay benchmark spanning 1,500 users, FSPO + RAT attains a 90.3% win rate, within 0.6 percentage points of prompting with the oracle, ground‑truth persona (90.9%), showing that the generated descriptions reliably recover the effective user profile needed for generation. On Reviews, adding RAT to 8‑shot FSPO increases win rate from 80.4% to 92.3% on trained users (84.6% on interpolated users), again indicating that the two‑stage procedure is robust and beneficial even when the description is model‑generated rather than provided. Appendix A.2.1 further illustrates that RAT descriptions qualitatively match the hidden persona used to generate preferences, consistent with the training objective that prefers semantically aligned descriptions. These results collectively support the claim that RAT is robust in practice and that small imperfections in the textual description do not materially harm performance because the model is concurrently conditioned on the original few‑shot preferences that generated RAT in the first place.

---

> > ### Author Response · Authors · 2025-11-17
> > **Response to Reviewer 5VkS [2/N]**
> >
> > > Since the setup is preference-style, are judges from the same model family? Are they sensitive to verbosity/style
> >
> > The LLM judge used in our evaluation (GPT-4o) is a more capable model than the learned model under assessment (Llama 3.2 3B Instruct). For synthetic evaluations, we strictly follow the AlpacaEval 2.0 framework [2], thereby leveraging its established robustness and methodological rigor. As described in Appendix A.3, we applied the same level of diligence and consistency when conducting evaluations with real users. We reiterate the setup below.
> >
> > To minimize bias, we normalized both FSPO and baseline responses across three key dimensions. First, since response length can influence judgments, we computed the average length of FSPO outputs and instructed the base model during evaluation to generate responses close to this average (approximately 250 words). The SFT baseline naturally aligned with this since it was fine-tuned on the same preference dataset.
> >
> > Second, to maintain consistency and fairness given the context-length limits and instruction-following capabilities of smaller open-source models, we standardized the response format to paragraphs rather than markdown, particularly for the Roleplay task. The base model was explicitly prompted to follow this format.
> >
> > Finally, differences in the number of views presented can skew user preferences, as users often favor concise and direct answers. To address this, we ensured that any pair of responses compared contained the same number of views. Moreover, responses were presented in random order to eliminate positional bias and ensure a fair evaluation.
> >
> > **Please let us know if you find your concerns addressed, and if so, we would be grateful if you are willing to raise your score.**
> >
> > References:
> > 1. Generative Verifiers: Reward Modeling as Next-Token Prediction
> > 2. Length-Controlled AlpacaEval: A Simple Way to Debias Automatic Evaluators

---

> > > ### Author Response · Authors · 2025-11-26
> > >
> > > Dear Reviewer,
> > >
> > > Since there is only 1 week left in the rebuttal period, it would be helpful for us to know if our rebuttal above addressed all your questions and concerns, or if any others remain. We are happy to discuss further and provide more evidence if it is helpful. If all your concerns are addressed, we would be grateful if you could acknowledge that soon.
> > >
> > > Thanks,
> > > Authors

---

### Official Review · Reviewer_PZpS · 2025-11-03

**Soundness:** 3
**Presentation:** 2
**Contribution:** 2
**Rating:** 6
**Confidence:** 4

**Summary:**

The paper proposes FSPO, a meta-learning approach for LLM personalization. FSPO uses a few user-specific preference examples to quickly infer a personalized reward function for a user, potentially with a user description rationalization (RAT) step that in addition infers a natural language persona summary to further improve personalized generation. This paper also constructs  a large synthetic preference dataset consisting of 1M synthetic personalized preferences. Experiments across three tasks and a real human study shows that FSPO outperforms standard prompting and SFT.

**Strengths:**

1. Conceptually sound meta-learning framing with RAT, which is practical and interpretable
2. Well-designed synthetic data pipeline with large scale dataset as a result
4. Actual human study

**Weaknesses:**

1.A core desideratum for personalization is interactive, continual adaptation: as the LLM interacts with the user, it should actively infer, refine, and remember their preferences in context. Therefore, one major concern is over the fixed k-shot evaluation missing the crucial dynamic evolution. Specifically, there are a few dimensions to this
i. when the models are trained on k-shot optimization, but at test time, the user interactions are not k, but either less than or more than k.
ii. the dynamic of how well the model does in terms of continually learning user preference as the model interacts with the user further, i.e., it might be the case that the model trained is better at eliciting preferences when interaction is sparse, but fails to remember well when interaction gets longer compared to context-engineering at inference time only (no training). It is informative to provide how the model performs across interaction turns.
iii. metric could be some sort of a regret-style curve over turns (win-rate or utility vs. step), not a single snapshot. We want to know how the performance scales from under-conditioned to over-conditioned regimes. Ideally, adaptation is monotone and data-efficient: small k yields robust gains, larger ks keeps helping without overfitting or context bloat.

2. Out-of-distribution: conversations naturally shift topics. It'd be informative to see evaluation over topics that are not seen during training and how good the model is at eliciting preferences given ood interaction histories.

**Questions:**

1. What if we are at a cold start problem? How is the model trained to pick up preference in a most efficient way.
2. The few-shot interaction that is provided, what if you have the chance to choose the questions to ask? Is there discussion over active preference sampling. The assistant should really selectively elicit the next most informative pair/question to shrink uncertainty in the user latent with minimal burden.

---

> ### Author Response · Authors · 2025-11-17
> **Response to Reviewer PZpS [1/N]**
>
> We thank the reviewer for their insightful comments and feedback, particularly for acknowledging the efficacy of our meta-learning framework for personalization and the design of our synthetic data generation method.
>
> > Out-of-distribution: conversations naturally shift topics. Topics that are not seen during training and how good the model is at eliciting preferences given ood interaction histories.
>
> In the roleplay task, we explicitly investigate this capability. For this domain, we evaluate the model exclusively on held-out users (personas), previous user interactions (preference histories), and novel questions. This setup allows us to rigorously assess the generalization ability of FSPO. Importantly, the users and their underlying preferences are completely unseen during training, ensuring a true test of OOD robustness.
>
> Moreover, our results demonstrate that this generalization extends to real-world users, whose preferences naturally differ from the synthetic ones used in training. This highlights FSPO’s potential to handle diverse, unseen conversational patterns and effectively adapt to out-of-distribution user behaviors encountered in realistic deployment settings.
>
> > fixed k-shot evaluation missing the crucial dynamic evolution
>
> This is a great question! In Appendix A.1.1 (Table 7, right), we observe a monotonic increase in model performance with the number of few-shot examples in the Roleplay dataset. This trend suggests that as additional contextual examples are provided, the model’s task performance consistently improves. These findings offer preliminary evidence that model adaptation is both monotonic and data-efficient, benefiting proportionally from more user-provided demonstrations. This is copied below for your convenience:
>
>
> | Few-Shot Examples | Winrate (%)     |
> |--------------------|----------------|
> | 1                  | 65.7           |
> | 2                  | 69.3           |
> | 4                  | 72.1           |
> | 8                  | 82.6 (reported)|
>
>
> In scenarios where more examples are available than the fixed shot limit, we find that random subsampling provides a competitive baseline. Using the ELIX-Hard dataset, we further evaluate an active sampling strategy that employs a retriever conditioned on the query to select the most relevant subset of examples. Specifically, we use the `intfloat/e5-mistral7B-instruct` model to retrieve examples from a set of double the preferences for a particular user. Our results show negligible performance differences compared to the standard 4-shot configuration:
>
> | Method              | WinRate |
> |----------------------|---------|
> | 4 shot (reported)    | 91.8    |
> | 8 shot (subsampled)  | 91.9    |
> | 8 shot (retrieved)   | 91.7    |
>
> However, one unexplored aspect is the scenario where the number of evaluation shots is less than the number used during training. Following PaliX[1], both training and evaluation could incorporate a dynamic shot count, with mechanisms such as Attention Reweighting or Retrieval-based Context Selection to manage larger contexts effectively (see Figure 6). Adopting such a strategy for FSPO could enhance adaptability to user-specific data. We leave this analysis to future work. That said, FSPO only requires 4–16 preference labels per user, which are efficient to collect, averaging between 15-20 minutes in our user study. This short “burn-in” period is consistent with common practices in recommender systems, as noted in [2].
>
> > A core desideratum for personalization is interactive, continual adaptation: as the LLM interacts with the user, it should actively infer, refine, and remember their preferences in context. Is there discussion over active preference sampling?
>
> This is an excellent point, and we wholeheartedly agree! We believe that FSPO is a step towards this where the model is able to take in experience collected from a user to further align with their preferences. An ideal LLM should dynamically infer, refine, and consolidate these preferences through ongoing interaction. As the model engages with the user across contexts, it should continuously update its internal representation of the user’s goals, values, and communication style—thereby enabling more coherent, contextually aligned, and user-specific behavior over time.
>
> We recognize this as an important direction and will include an expanded discussion in the Future Work section, with additional implementation considerations outlined in the Appendix.

---

> > ### Author Response · Authors · 2025-11-17
> > **Response to Reviewer PZpS [2/N]**
> >
> > > Cold start problem
> >
> > This is a great question. We think that this is a strength of the Synthetic Construction approach proposed in this work and have outlined this in Appendix A.11. We restate the argument below:
> > One advantage of the synthetic construction proposed is addressing the cold-start problem. When launching a new personalized feature, there is often no historical data to draw upon. FSPO provides a robust initial data curation pipeline that can deliver immediate value, as evidenced in the tasks studied in this work. This extends to situations where an organization has a wealth of user data, but not in a format amenable to LLMs (e.g, a housing and neighborhood commerce network such as BILT[3], which has a set of user transaction patterns and platform engagement not standard to LLMs). In such instances, a synthetic preference dataset can be designed using FSPO, based on the existing data signals. Additionally, as real preference data is collected, it can be integrated with or used to fine-tune the synthetically trained model, demonstrating how FSPO can serve as a critical foundation and accelerator. Furthermore, works such as AlpacaFarm[4] have been introduced for prototyping/development of preference-based systems. As stated in their abstract, synthetic data such as LLM Prompts can simulate human feedback that is 50x cheaper than crowdworkers and display high agreement with humans (corroborated with our human study). Thus, in many real-world applications, this synthetic data generation pipeline can be used for benchmarking purposes that emulate a more realistic downstream application in personalization.
> >
> > **Please let us know if you find your concerns addressed, and if so, we would be grateful if you are willing to raise your score.**
> >
> > References:
> > 1. PaLI-X: On Scaling up a Multilingual Vision and Language Model
> > 2.  Cold-start Recommendation by Personalized Embedding Region Elicitation
> > 3.  How Bilt Built a Million-Agent Recommendation System with Letta
> > 4. AlpacaFarm: A Simulation Framework for Methods that Learn from Human Feedback

---

> > > ### Author Response · Authors · 2025-11-26
> > >
> > > Dear Reviewer,
> > >
> > > Since there is only 1 week left in the rebuttal period, it would be helpful for us to know if our rebuttal above addressed all your questions and concerns, or if any others remain. We are happy to discuss further and provide more evidence if it is helpful. If all your concerns are addressed, we would be grateful if you could acknowledge that soon.
> > >
> > > Thanks,
> > > Authors

---

### Author Response · Authors · 2025-12-01
**Rebuttal-Period Summary – Key Clarifications & New Results**

We thank the reviewers and the AC for their thoughtful and constructive feedback during the rebuttal period. The discussion has materially improved the paper’s clarity, empirical grounding, and positioning within the literature. For the convenience of the new AC, in light of the recent ICLR policy change, we summarize these updates below:

- **k-Shot & Dynamic Evolution**: We provide analysis showing win rates increase monotonically with the number of few-shot examples (65.7%, 69.3%, 72.1%, 82.6%) in the Appendix. For settings where a larger number of preferences are provided, both random subsampling and retrieval-based selection are viable approaches, as seen in the new active sampling experiment.

- **OOD Generalization**: We clarify that the Roleplay evaluation uses held-out personas, histories, and queries. This demonstrates that FSPO generalizes to unseen users rather than overfitting to fixed preferences. The real-user study also showcases the robustness of our pipeline to OOD personas.

- **Addressing Cold-Start**: We emphasize that FSPO’s synthetic pipeline directly targets the cold-start problem, in settings with limited or no historical preference data.

- **Quantified Sim2Real Transfer**: Human evaluation confirms a ~70% win rate. A new ablation shows that removing View Conditioning and Iterative Improvement drops the win rate to 56% (near random), underscoring the contribution of structure and diversity in our preference data construction process.

- **Inference Cost**: We now include an analysis of the inference overhead of RAT (avg. 239 tokens) compared to generation (896 tokens). This cost is unlikely to be a bottleneck, especially for reasoning-focused LLMs, and can be amortized through prompt caching.

These additions and clarifications address the major concerns raised during the review process. We respectfully request that the AC consider these updates in their final evaluations.

---

### Meta-Review · Area_Chair_7a5x · 2026-01-06

**Summary:**

This work proposes a few-shot preference optimization (FSPO) algorithm for LLM personalization and designs a preference data synthesis method. Experiments verifies the effectiveness of FSPO. The reviewers are generally positive about the FSPO framework and synthetic data pipeline.

**Reviewer Concerns:**

Reviewers had questions regarding specific details of the experiments. Major concerns include:
- The sim2real transfer is not sufficiently quantified.
- It is unclear how robust RAT is when user descriptions are partially or fully incorrect.
- Evaluation across interaction turns is missing.
- It is unclear how well the model elicits preferences on topics not seen during training.

The authors address most of those concerns during the rebuttal. Results on sim2real transfer, k-shot and dynamic evolution, OOD generalization are provided. For RAT robustness, they argue that controlled corruption experiments may conflate robustness with generalization. Instead, they show that RAT is robust to minor description errors.

**Reviewer Scores:**

The original scores were 6/6/4. The reviewers were likely to maintain or increase their scores.

---

### Decision · Program_Chairs · 2026-01-26

Accept (Poster)